# The Role of Social Network Analysis in Social Media Research

**Zhou Nie *** **, Moniza Waheed** ⓘ **, Diyana Kasimon and Wan Anita Binti Wan Abas**

Department of Communication, University Putra Malaysia, Selangor 43400, Malaysia;
moniza@upm.edu.my (M.W.); diyana.nawar@upm.edu.my (D.K.); anita@upm.edu.my (W.A.B.W.A.)
* Correspondence: gs46697@students.upm.edu.my

**Abstract:** Previous studies regarding social interactions commonly adopt research methods that investigate causal relationships between variables. The existing approaches often utilize variables derived from general contexts, aiming to apply them universally across diverse situations. However, social interactions, including the usage of social media, are intricately woven within the immediate social context. The interpretability of these generalized variables has been attenuated by the dynamic and transient nature of social contexts; these variables have diverse impacts on social interactions. Consequently, researchers have been diligently seeking new variables relevant to specific social contexts in order to complement the existing generalized ones. However, the ever-changing nature of social contexts poses a challenge, impeding researchers from exhaustively defining all variables that influence social interactions. To address this complexity, this study proposes social network analysis as a suitable research method capable of capturing the ever-evolving dynamics of social interactions, including social media usage. Furthermore, this study puts forth hypotheses that specifically explore the role of individual social networks in social media research, with the aim of stimulating future investigations that center on the interactive and dynamic nature of social media usage.

**Keywords:** social network analysis; interactions; social media usage; social contexts; communication theories

## 1. Introduction

Social media is the most extensively utilized type of communication in the world because of its potent ability to digitalize various forms of communication, from interpersonal communication to mass communication. According to a worldwide investigative report, users spend an average of 6 h and 35 min on social media, with connections with others being the most popular purpose for social media usage [1]. By April 2023, there were 4.80 billion social media users, which constitutes 60% of the world's population [1].

Although digital technologies allow for virtually any form of communication, including text, video, and synchronous or asynchronous communication, people's social media communication strategies are still influenced by the social contexts in which they are incorporated. According to earlier research [2–4], people utilize social media differently depending on their social and cultural backgrounds. Social media appears to impact the shallow levels of behavior, and it has little influence on the deep levels of behavior; rather, it serves as a recorder of the different interactions that individuals engage in in real life.

The primary explanation might be that a combination of factors from their ingrained beliefs and the current social contexts shape their behaviors. Hofstede [2] made the observation that people's behaviors are heavily influenced by their cultural beliefs, which are developed daily through their growth. Cultural beliefs are thus implanted at a deeper level of consciousness, where newly acquired knowledge and short-term memory have little impact. On the other hand, Granovetter [3] pointed out that, aside from situations where behavior is influenced by various cultural beliefs and values, people's behaviors are also influenced by the people around them. In his theory of threshold, he gives the following example: if a person's threshold for doing a behavior is 5, he or she is less likely

to perform such a behavior in a group of people with thresholds between 1 and 4 than in a group of people with thresholds beyond 5. To put it another way, a person's behavior may be stochastic in some scenarios in which the general explanation of behavior is prevented from its usual sway.

As a result, we can see the research challenge at hand: social behaviors are formed by a combination of individual and social factors, which are dynamic and transient in different cultural backgrounds and social contexts. Scholars typically find that while some factors have strong capacities to explain behavior in some contexts, these capacities were significantly attenuated in other circumstances, meaning that the factors could not always account for behaviors in all circumstances [3,5,6]. The recent research methodology commonly emphasizes the utilization of normal distribution of data, but its applicability is constrained to stable and general scenarios. Social network analysis, however, is a fit method for studying dynamic and transient social contexts. In social network analysis, individuals or other entities are represented as points, and the relationships between these points are represented as lines. Points and lines in network graphs can then be used to display the structures of dynamic interaction. This geometric simplification could make structural characteristics like density, weight, and heterogeneity analyzable by mathematic approaches, which could reveal the mechanism behind dynamic social behaviors more succinctly.

According to Knock and Yang [7], social media usage consists of interactions between users, including both individuals and other entities. Observing different types of interactions offers insight into how interactions form social networks, which bridges micro perspectives and macro perspectives regarding social interactions, as well as the structural characteristics of social networks formed by different types of interactions, such as strong or weak ties that connect individuals within social networks, centrality structures, which refers to individuals inside a social network having the tendency to act surrounding one center, transition structures, which means that people inside one social network tend to expand their connectors outside their social network, etc. The method of analyzing the relationships between individuals or other entities within a social system is called social network analysis (SNA) [3,5]. With the use of social network analysis, scholars will be able to conduct social media research from a dynamic and systematic approach that may be more reflective of reality. This paper is a literature review paper including mainly six sections, which are introduction, significance of this study, related works on social media usage, related works on social network analysis, possible hypotheses, and conclusions. This construction aims to look for the possibility of combining social media research and social networks analysis, as well as proposing possible hypotheses frameworks for future studies.

## 2. The Significance of This Study

The most important significance of this study is to introduce social network analysis into the studies of social media usage and to bring a paradigm shift in social media research, which emphasizes that the systematic and relative perspective should be the proper research method for dynamic social interactions. Social network analysis, which investigates online interactions on a highly abstract and simplified level, enables researchers to measure the complex process of social media usage in dynamic and transient social contexts. The academic and practical significance of the introduction of social network analysis into social network analysis is introduced in the following sections.

In the aspect of academics, social network analysis may result in a paradigm shift in research methodology regarding social media usage. Regression analysis for relationships between factors is universally used in the empirical studies on social media usage that have been published in the last decade [6,7]. However, because these factors were derived from particular social contexts, they might have different effects on social media usage in other social contexts. For instance, the variable of attitude could have more effect on behaviors in the culture of individualism than in the culture of collectivism because the collective culture values group goals more than individual goals, which makes individual attitude less important than in individualism culture [2,4,6]. Because social media use is

the result of the interaction of a series of intricate social processes, it is nearly impossible to identify all the factors that participate in these social processes. As a result, scholars must constantly elicit new factors to adjust to transient contexts, and these new factors might not have the same interpretation capabilities when social processes change, which wastes effort to an extent.

Social network analysis adopts a systematic approach to capture the transient features of social media usage by focusing on the outcomes of interaction rather than precedent factors of interaction, which contrasts with the method of constantly creating new factors to explain social media usage in various social contexts. The shift of research focus may save researchers from defining additional, occasionally semantic overlapping factors and aid in their better intuitive understanding of the entire social media usage process.

With regard to the aspect of practical significance, social network analysis could show how various groups of people use social media in various social contexts, which could be useful to those who are interested in the spread of information and influence, the dissemination of innovations, or the effects of advertising. There is a survey that claims that 65.2% of online users receive information about their social lives on social media from their networks, making network interactions on social media one of the most significant loci of social media usage [1,8,9]. Therefore, understanding the mechanism of interactions on social media could help people from the corporate, public administration, and other sectors to more accurately comprehend the effects of using social media.

An overview of the literature on social network analysis and studies of social media usage is given in the following sections of this article. To enable a shift in research methodology for the study of social media usage, the goal of this review paper is to evaluate the possibility of integrating social network analysis into social media studies. For discussion, the potential hypotheses will also be given.

### 3. Works Related to Social Media Usage

Recently, the world has increasingly used social media for communication. The focus of the definition of social media is typically on interactive internet-based and digital-based technologies [8]. Through functions offered by websites and applications of social media organizations, people can digitalize information, ideas, and other expressions and then send these contents via the internet to any other user. On social media, users can decide to create their own networks for various purposes [10–12].

In the past decade, there has been a substantial increase in research pertaining to social media usage. Researchers have employed various theoretical frameworks to gain insights into this phenomenon, primarily encompassing personal behavior theories, social behavior theories, and mass communication theories. Personal behavior theories [7,13,14], exemplified by the theory of planned behavior, focus on the influence of individual traits, with intention being a critical determinant of social media usage. On the other hand, social behavior theories [15,16] delve into social variables that either facilitate or impede human behaviors, exemplified by the social loafing theory, which highlights the differences in behavioral patterns between individuals in collective groups and individual settings. Mass communication theory [17,18] emphasizes the interaction between individuals and social media, with particular emphasis on the causal relationship between social media usage and its impact on individuals. This review aims to explore and introduce various perspectives in social media research beyond these established theories.

As an innovative way to connect people globally online, researchers are interested in learning how much social media use can alter users' behaviors and whether it has the potential to be a democratic force in the formation of network societies [19–21]. However, as researcher Castells [4] has noted, social media could not significantly alter current societies if it were not publicly owned. In addition, since all content created by users on social media platforms could be traded for profit, social media use is merely an extension of industrial reproduction at this stage.

Regarding the cultural aspect, cultural beliefs are a significant factor in impacting social media usage. According to Hofstede [2], cultural beliefs operate on the deep level of consciousness where affections and personality are developed through daily interactions with the environment as a person grows, much like the formation of conditioned reflexes. As opposed to the deep level of consciousness, which generates emotional thinking, the intellectual level, or upper level of consciousness, develops reasoned thinking, which requires more time and focused attention to make decisions. The primary explanation may be that the regions of the human brain that control the conditioned reflex and rational thought are different [9,10]. When people deal with unfamiliar information, such as learning new knowledge, human rationalities may be at play. However, when people are in familiar contexts, they tend to rely on their well-established experiences, which is why most of the social media usage regarding common social lives is localized, and usage patterns vary across different demographic groups [11,12].

Social media use is an intermingled social process that includes intellectual and emotional aspects; the aspect that takes effect depends on the social context. Because social contexts are transient and dynamic, it is difficult to determine the universal causes of social media usage behavior by deducing from the diverse results. When examining the research on social media use, most of the studies look for new factors to explain social media use based on established models such as the theory of reasoned action [22], the theory of planned behavior [23], and other mass communication or behavioral theories [23,24]. These theories discuss the relationships between individual factors and behaviors. Regression correlations between these variables, however, could only have examined a predetermined set of assumptions, which may only partially explain social media usage. The likelihood of departing from the real scenarios increases as more focus is placed on attempting to find elemental and stable factors that determine behaviors, and the interaction between behaviors and social contexts is neglected.

Looking for the elemental constituents of matter is a common research method in natural science. Natural science researchers generally seek to find the stable and valid composition of matter and try to repeat the findings. This approach calls for environmental factors to be carefully controlled, which may not be feasible when it comes to social events. The environmental conditions are dynamic and stochastic when real social phenomena occur, thus the models derived from controlled conditions might not be accurate enough to represent actual situations. People may behave differently when they are in contexts with outside factors controlled and when they are under real social conditions, according to Granovetter [3].

A great number of examples from studies on social media usage demonstrate how difficult it is to define all factors that influence social media use in a variety of social circumstances [19–21]. According to empirical studies, social psychology's and other behavioral theories' hypotheses can only explain a restricted range of behaviors, and their interpretation varies with the social contexts. The individual factor of attitude, for example, has a big impact on how social media users decide which green products to buy, whereas intimate relationship communication with a collective cultural background on social media has lessened its impact [13,14]. The basic cause of the volatility of these factors is the diverse and dynamic nature of the social contexts used to infer these factors.

More specifically, the theory of planned behavior has been extensively applied in studies of social media usage. There are three boundaries for application of the hypothesized model, which were mentioned by Fishbein and Ajzen in 1975 [22,23]. They are personal control, behavior stability, and behavior specificity. These boundaries have defined the scopes within which hypothesized variables could have a significant impact on behavior. However, scholars have still been striving to use it outside of these three boundaries by adding more significant factors to the model in order to strengthen the original model's interpretive flexibility and ability to adapt to various social contexts. They consequently discovered that the variables in the original model had various effects on behaviors; for example, attitude is significant in the prediction of an athlete's moral behaviors, but the

impact of subjective norms is relatively minor [15,16]. It became clear that the stability of the theory of planned behavior's hypothesized model was challenged. Thus, the paradox between the pursuit of stability of predictive models and dynamic social contexts has been raised.

When considering research methodologies, linear regression is a commonly employed approach to investigate relationships among hypothesized factors [24]. Nonetheless, the utilization of linear regression analysis encounters certain limitations when applied to real-world data [6,24]. Firstly, the validity of analysis results can be compromised when attempting to apply linear regression to non-normally distributed data, as deviations from normality can substantially influence means and standard deviations. Secondly, the exploration of relationships within small groups is hindered by the central limit theorem, which necessitates a large volume of data to confirm normality. Consequently, the investigative capacity of linear regression is restricted in such cases. Thirdly, in real social contexts, factors exhibit interactions and dependencies, whereas linear regression assumes the independence of factor data, thus potentially overlooking crucial aspects of factor characteristics. These limitations underscore the need for alternative methodologies that can address the complexities and inherent nuances of real data.

The factors that affect social media usage are impossible to tell apart and often overlap. According to Ajzen [17], the notion of perceived behavioral control and the concept of self-efficacy, or locus of control, may be tautological in the theory of planned behavior. It is challenging to draw clear distinctions between these related concepts. In reality, the behaviors we observe are the results of a variety of factors intermingled, similar to a cup of black water, and we are unable to tell which colors were added to the water because they were already mixed together. We cannot clearly tell what the precedent decisive factors are because this process is irreversible, comparable to reversible scientific experiments performed under controlled conditions. As a result, when researchers attempt to retrieve concepts from correspondents' reports, they usually discover that many concepts overlap and are blurred in the memory of the correspondents [25]. Correspondents are only able to remember some general feelings rather than every specific detail of behavior because the consciousness of the human brain is only capable of memorizing a limited amount of information. As a result, Ajzen [18] opined that when it comes to human behaviors, it is preferable to take into consideration the degree of behavioral performance rather than specific kinds of impacted factors of behaviors.

It is insightful of Ajzen [17] to bring up the simplified method of analyzing human behaviors. This method might exempt researchers from endlessly defining factors that influence social media studies across transient social contexts. The focus of many scholars has shifted from developing factors that govern social media usage to the features of social media usage presented in real circumstances. According to a survey on global social media usage [1,26], connection with others is the primary use of social media across different cultures. Through its practical and potent features, social media has so far mostly fostered connections between people, drastically reducing the time-space barriers in communication. However, varied cultural backgrounds and social contexts determine how people utilize social media functions and what those functions denote. According to researcher Burke [19], diverse uses of social media functions could lead to various connections.

The use of social media can be roughly categorized into three types: direct interaction with targeted individuals, passive consumption, and broadcasting [19]. The term "direct interaction" refers to interactions with specific users on social media that are intended to maintain existing relationships or create direct connections between users. The most common functions of social media in direct communication include the like button, inline comments, messages, synchronized chat, and photo tagging. On the other hand, interactions with an unspecific number of contacts are referred to as indirect communication. Indirect communication, which includes the utilization of the aggregate stream of news, status updates, links, and profiles on social media, includes passive consumption and broadcasting. Direct communication requires more concentration and attention than indi-

rect communication, because maintaining relationships with concrete individuals needs more supportive and caring feedback to the targeted individuals, so generally speaking, direct communication has a higher propensity to create strong bonds between people. while indirect communication has a greater chance of forming weak ties or bridging ties, through which individuals could reach more unspecific contacts.

However, as the number of empirical investigations grew, researchers discovered that strong ties don't always imply bonding relationships, and weak ties don't always indicate bridging effects [3,10,17,18]. For instance, when it comes to intimate relationships, weak ties may often exist in those individuals who believe that public displays of affection are more appropriate. Concerning social media usage, those people are more likely to have indirect connections like wall posting, status updates, and profile links, which are considered better ways to facilitate their intimate relationships. On the other hand, when people place high importance on privacy in intimate relationships on social media, they may plan to use more direct communication features, like messages and synchronized chats. Depending on the general agreement over the behaviors that constitute intimate relationships, people may choose different types of ties [19]. The consensus varies with social contexts. Therefore, just as Granovetter [3] reminded us, the paradoxes that empirical studies ran into just exhibit the complexity and transiency of human behaviors.

## 4. The Related Works of Social Network Analysis

The research methodology for social media usage should be able to capture the dynamic interactions between social media usage and social contexts since human behaviors, including the use of social media, are susceptible to complex social situations [21,22]. Social network analysis may be one of the approaches that are best suited to capture these interactions. According to its definition, social network analysis (SNA) is the process of analyzing interaction structures by designating interacting objects, such as people or other entities, as nodes and the relationships between them as lines or edges [3,5]. Using this technique, network graphs made up of nodes and lines can be drawn, and the structures can then be institutionally presented.

The interaction between the micro level, which includes individuals, and the macro level, which includes various social groups, could be captured through social network analysis since connections on both levels are made into visible graphs. It is a research method that investigates both the micro and macro levels of human behavior, making it potentially more reflective of actual human behavior. Because most human behaviors always take place among interactions between micro and macro levels, humans learn about the outside world and adjust their behaviors through interaction and exchange with the outside environment; therefore, understanding the interactions between individuals and their environments is crucial [3,5,20]. Although a little abstract, it is clear that if individuals act independently and without any interactions, they may be free to act however they like, making their behavioral patterns elusive and unpredictable. However, when acting in social groups, an individual's behavior is governed by the disciplines present in these groups because the freedom an individual has is constrained by groups. These laws resemble the entropy laws of physics [23].

Entropy is a fundamental concept utilized to gauge the level of uncertainty or randomness pertaining to the outcomes of a random variable [23–25,27]. It provides a quantitative measure of the average information necessary to describe the various possible states or forms exhibited by a factor. Essentially, the number of potential manifestations of a random factor X is contingent upon the entropy value of the system in which it operates. The explicit formula representing entropy is presented below:

$$H(x) = -\Sigma P(x) \times \log_2 P(x) \qquad (1)$$

Formula (1) introduces the concept of entropy [24], where $H(x)$ denotes the entropy of a random variable X. $P(x)$ represents the probability of a specific outcome or event x, emphasizing the likelihood of observing that particular outcome. The symbol $\Sigma$ denotes the

summation operation, requiring the subsequent expression to be summed over all possible outcomes x of the random variable X. The logarithm base 2, denoted as $\log_2 P(x)$, scales the probabilities logarithmically and serves as a measure of information. By multiplying P(x) by $\log_2 P(x)$, we obtain the information content associated with each outcome, quantifying the number of bits needed to transmit the outcome x. Furthermore, the negation of the sum of these values $(-\Sigma P(x) * \log_2 P(x))$ ensures that the entropy is expressed as a positive value, effectively capturing the average amount of information required to describe the random variable X.

The basic logic of entropy is widely employed in statistical analysis [28–30]. Probability serves as a pivotal concept that captures the interplay among factors within a specific system, acting as a bridge between micro-level and macro-level phenomena. Researchers have observed that isolated individual particles demonstrate stochastic and unpredictable behaviors. However, when multiple particles interact, their collective behaviors manifest discernible and recurring patterns, enabling estimation of the probabilities associated with each pattern. The patterns of a single particle are unpredictable because single particles exhibit equal probabilities for all potential behavioral patterns, just like the wave-particle duality. The emergence of specific patterns becomes prioritized when particles interact and integrate into a group in order to fight against disorder and disintegration. Consequently, these prioritized behavioral patterns acquire different probabilities of observation. The ability of a system or group to maintain itself is contingent upon the entropy of the system, representing the multitude of observable patterns within the group or system [24].

Similarly, individuals obtain prioritized behavioral patterns through interactions with their environments. These dynamic interactions between individuals and their environments can be easily observed because of the technique of digitization and the internet [4,12,21]. Digitalization refers to the process of using digital technologies to transform and optimize various aspects of business operations, services, or processes [12,31]. This transformation entails the utilization of two distinct states of electricity, namely flow and stop, to encode and process information within a simplified system. The primary objective of digitalization is to achieve enhanced efficiency and accuracy, resulting in a reduced probability of errors or false outcomes. Through the binary system, which consists of different combinations of 0 and 1, people can create multi-channels of content including video, audio, and visual information that can be converted directly into electric impulses, namely, flow and stop. On the other hand, internet-based methods can deliver this multi-channel information to any receivers, including computers, iPads, and smartphones. Communication between people is substantially facilitated by digitalization and the internet. Theoretically, on social media, people may connect without any time-spatial barriers. Therefore, communication on social media is a process that can integrate interpersonal and mass communication, as well as micro and macro levels of interaction.

Digitalization codes information into digital signals to make human communication readable by electrical devices; similarly, social network analysis codes interaction into graphs made up of nodes and edges [5,29,30]. Just as the capabilities of digitalization can encompass various kinds of communication, social network analysis can also include the diverse processes of interaction between individuals and other entities. By transforming individuals and other entities into geometric units such as nodes, as well as the lines that present the relationships between these nodes, the basic units of human behaviors that are interactions could be represented by a set of two nodes and one line between them. With the simplified analysis units, the complex social processes of interactions between micro and macro levels can be analyzed through geometric approaches, which are good at explaining the dynamic changes of structures. Thus, social network analysis could be a fit approach to study the behavior of social media usage; hence, social media usage can be seen as the result of interaction between individuals and online environments.

Just like the basic logic of entropy, social network analysis estimates the probabilities of basic units of interactions through exponential random graph models, as Equation (2) presents:

$$\Pr(Y = y) = \left(\frac{1}{k}\right) \exp\left\{\sum_A \eta_A g_A(y)\right\} \tag{2}$$

In Equation (2), the variable $Y$ encompasses the entire set of potential ties within a fixed social network comprising n actors. For instance, in a social network with five actors, the sum of all possible ties among them amounts to 10, determined by the combination formula $C^5_2 = 5!/(2! * (5 - 2)!) = 120/12 = 10$. In this context, the variable $y$ represents the observed ties within $Y$. The tie density, denoted as $g_A(y)$, refers to the ratio of observed ties to all possible ties within the exponential random graph model. Here, the term $A$ represents a specific structural feature or substructure present in $Y$. $A$ serves as an identifier for the research-oriented structural units within the specified social network, as demonstrated in Table 1. Table 1 provides an illustration of how a research unit, referred to as an "edge," encompasses two actors and the tie between them. It further depicts how a structural feature consisting of three actors can be classified as a two-star unit or a triangle unit. Additionally, it exemplifies how a three-star feature involves three actors surrounding a central actor.

**Table 1.** Structural features in social networks.

| Parameter | Structural Features | Estimate (Standard Error) |
|---|---|---|
| θ (Edge) | | −3.12 (1.36) |
| σ2 (Two Stars) | | 0.06 (1.84) |
| σ3 (Tree Stars) | | −0.02 (0.13) |
| τ (Triangle) | | 1.06 (8.4) |

To be specific in the research approach, the probabilities of structural features inside a certain social network, or a system, can be estimated via social network analysis [25,26]. The primary technique, known as the Monte Carlo maximum likelihood estimate, uses the data gathered to estimate the probabilities of each structure inside a system. Monte Carlo maximum likelihood estimation can be particularly useful in complex statistical models where the likelihood function cannot be easily solved analytically [27,30,31]. Simulating data from the model and computing the likelihood over these simulated samples provides an approximation of the true maximum likelihood estimation. The estimated results are more accurate when more information about structural features is provided.

Table 1 shows an example of a social network made up of 28 students [25,27]. An edge, which has at least two nodes and one relationship between them, is the basic analysis unit of a structure. The essential value for the estimation of parameters that show the probabilities of various structural features is the density of edges in a social network. The density is the ratio of the observed number of edges in a social network to its maxim number. For instance, if a social network of five people has three edges observed, and five people can have a maxim number of 10 edges, then the density of this social network is 0.3.

The density of this social network consisting of 38 students in Table 1 is 0.09, which is very low when compared to the maximum density of 1. Different structural features in a social network might affect the network differently. The density of edges may indicate the social network's average degree of connectivity. The feature of two stars has a higher propensity to form a closed triangle, and the triangle indicates a social network's propensity to fragment into small cliques [25,28]. On the other hand, the three stars' features might

represent transitive relationships between parts of social networks that could connect these parts. Regarding the estimation results in Table 1, the positive estimation results demonstrate that in this social network, only the two-star and triangle structures are capable of influencing the network, which exhibits a tendency to form closed cliques rather than the cohesive social network presented by the three-star and edge structures.

Due to the fact that they are primarily seen in daily life, these structural characteristics are representative of human behaviors. Based on the collected data, these structural characteristics can be either independent of or related to one another. In addition, age, gender, and socioeconomic position are features of nodes that could be utilized as conditions, or exogenous factors to modify the probabilities of structural features. Concerning features of nodes in social networks, the homophily hypothesis of probabilities states that, for instance, individuals with similar features may be more likely to connect with one another [25,29]. In fact, the more detailed the data gathered from various social contexts, the more accurate the social network analysis could be in the estimation of probabilities of structural features.

Aside from the general situations, there are a lot of situations wherein human behaviors take place, stochastic and transient. In these contexts, the reliability of traditional models that tend to explain behaviors by institutionalized norms and values such as norms, preferences, and beliefs, is greatly attenuated [20,25,31]. This is due to the fact that human behavior does not always conform to institutionalized and agreed-upon norms. People with the same norms and values may behave differently in different groups when engaging in certain collective behaviors, such as innovation, rumor diffusion, voting, and strikes [3,4,9]. This is largely dependent upon the nature of the groups; when there are few people participating in collective behavior, people are more likely to stay away from it; On the other hand, when there are more people participating, the movement is more likely to spread. Thus, exploring the norms, preferences, motives, and beliefs of individuals provides only a necessary but not sufficient explanation of behavioral outcomes [3].

However, whether the circumstances are general or stochastic, from the connective perspective of social network analysis, the structural features might always be captured and analyzed since the interaction between entities—the fundamental building block of social behaviors—is the unit of analysis. Basic units of interaction construct all complex and dynamic social behaviors. As a result, mapping human interactions at various sizes enables scholars to comprehend how these units combine to form functional structures to influence behavior in various contexts. As long as the data are available, it also enables researchers to track and examine how structures develop over time.

Social network analysis has garnered significant attention in investigating the structural characteristics of social networks. Key elements, such as network density, centrality, dispersion, homogeneity, and heterogeneity, have been studied extensively [32–34]. Advancements in social network analysis techniques have led to the utilization of advanced models, including p-models, to explore the independence and dependence of substructures within social networks. These models offer valuable insights into the fundamental and vital structures embedded in social networks, thereby enhancing our understanding of dynamic processes governing their formation. The significance of advanced models lies in unraveling the complexities of social network dynamics [35–39].

The dynamic social network analysis method may be more accurate in reflecting actual circumstances. As shown in Figure 1 [30], which uses the investigation of individual social networks on the internet as an example, people may interact with others in different ways depending on the situation, and social network topologies may influence how people communicate with one another [40–44].

Structure A in Figure 1 resembles a butterfly, signifying that two distinct social groupings are connected by a single individual. Structure B shows a fragmented individual social network made up of distinct cliques. Structure C depicts an individual social network that resembles an onion and has a loose connection to social groups outside of its center. As the only bridge connecting other social groups in the butterfly structure, the individual has the most potential to influence their social network. On the other hand, when people are in

social networks that are loosely connected, they might be less affected by others and might have fewer opportunities to connect with more social groups.

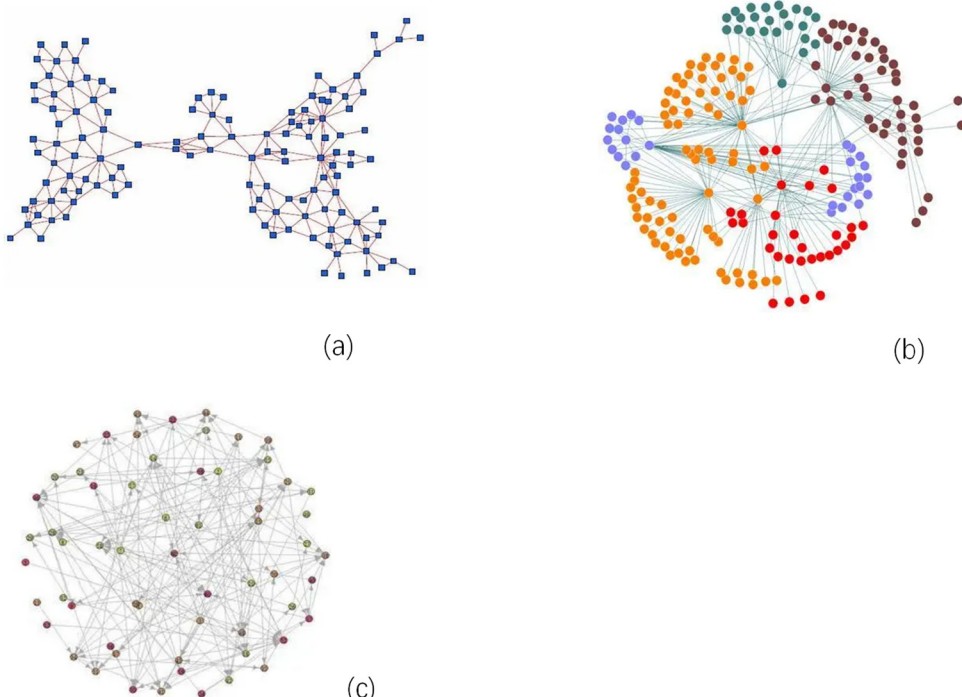

**Figure 1.** Different structures of individual social networks. (**a**) Butterfly structure. (**b**) Fregmented structure (**c**) Onion structure.

As seen in Figure 1, the positions of people in social networks could indicate their various behavioral patterns. People might be more relaxed and open when they are with family members in a social network, and they might act more formally when they are among co-workers. It's just like how identical particles acquire distinct behavioral patterns when they combine with various functional structures.

The primary goal of using social media is to establish connections with others [1]. One can't not communicate, according to Paul Watzlawick [31], and it is crucial to comprehend interaction and communication since they are the fundamental means by which people join social groupings and take on different social roles. Before social media, the investigation of interaction trailers between people and social groups was done through time- and resource-constrained field surveys and interviews. Social media could easily capture interactions between people and other social entities in digital formats, and the range of contact could be expanded to almost all social media users. The internet and digitalization techniques provide social network analysis with useful tools for examining the structural features of interactions on social media.

## 5. Possible Hypotheses Regarding Structural Features of Social Media Usage

Social network analysis, as discussed in this paper, provides a means to investigate the structural features of interactions on social media. The dynamic analysis could track the changes in interaction processes and thence could make a good simulation of the real social processes. With the development of digital and internet techniques, interactions on social media could be recorded accurately, which could benefit scholars of social media research to use social network analysis to better understand the structures of interaction [45,46]. According to the literature review of social media usage and social network analysis, the concept of entropy serves as a quantification of uncertainty and disorder within a system [23,27]. It signifies that observed phenomena stem from intricately intertwined and interdependent factors, defying clear differentiation. These phenomena are characterized by

irreversibility and dynamism, rendering the recursive analysis of all contributors to human behavior unattainable. Consequently, within behavioral inquiries, the causal probabilities underlying behaviors can merely be approximated based on generalized experiences. This forms the fundamental premise of the statistical p-model method employed in social network analysis, employing the maximum likelihood estimation (MLE) technique to mitigate uncertainties inherent in observed data. In the context of MLE, one could consider the parameter estimation process as trying to reduce uncertainty in the model by finding the parameter values that make the observed data most probable. This process aligns with the idea of minimizing entropy because reducing uncertainty corresponds to increasing the probability of observing the actual data points. This method could be more reflective of the dynamic and complex systems in which the behaviors occur. Thus, the hypotheses proposed below generally use the p-model to estimate the probabilities of each structure within the observed social network, which could investigate the functional structures behind social media usage. In other words, this method could take a connective perspective to observe human behaviors, compared to the isolated and separated perspective to study human behaviors. These hypotheses brought up in this paper could be used as frameworks for future studies. More empirical outcomes should be gained in various social contexts. H1 concerns the direct interaction on social media such as likes, comments, and synchronized chat. H2 presents the passive interaction on social media such as streams of news. H3 could investigate the interaction of broadcasting on social media.

**H1:** *The structural features of edges/2-stars/3-stars/triangles are more likely to occur regarding direct interaction on social media in the given social network.*

**H2:** *The structural features of edges/2-stars/3-stars/triangles are more likely to occur regarding passive interaction on social media in the given social network.*

**H3:** *The structural features of edges/2-stars/3-stars/triangles are more likely to occur regarding broadcasting interaction on social media in the given social network.*

More than these three hypotheses may be assumed in practical applications about the probability distributions of various social network structural features. The specific hypotheses should be based on scholars' research objectives. The more precise the information gathered from social networks on social media, the more precisely the probability distributions could be modified. Through empirical investigations, the levels of interactions from the micro level of people to the macro level of social groups and other social entities can also be included.

## 6. Conclusions

Social network analysis has been introduced for the investigation of social media usage in this review paper. Human interactions are dynamic and complicated social processes; hence an appropriate approach for dynamic processes is needed to describe these variations [24,31]. Previous theories have frequently used institutionalized norms and values—necessary but insufficient—to describe interactions. However, stochastic and urgent events like strikes, rumor diffusions, and riots frequently occur in real life, and human interactions don't always adhere to general values and norms. People with the same values and norms might be compelled by these events to behave differently than their usual behavioral patterns. Therefore, instead of making assumptions about a straightforward relationship between collective results and individual values and norms, employing a dynamic and systematic perspective to study interactions is preferable [3,10,20].

Social media has given us the perfect platform to observe dynamic interactions, at both the micro and macro levels [14,21,25]. Social media has made it possible to research digital records of interactions by using interactions as analytical units and data, which allows social network analysis to use mathematical techniques to analyze dynamic interactions on social media. Structural features are crucial to behavioral studies because they are functional to behaviors. Different structures could help or hinder the performance of behaviors.

In the past, dominant theories have tended to use well-established and agreed-upon patterns to describe social behaviors, while structural features that have strong interpretive capabilities for stochastic and dynamic interactions have been mostly overlooked. The underdevelopment of systems for tracking and analyzing interactions on various scales was a major contributor to the neglect. The potential of social network analysis to analyze dynamic interactions, however, would be more and more apparent with the development of the internet and digitalization techniques, which requires additional empirical research to verify it.

The adoption of relational and interactive social network analysis perspectives may transform the research paradigm of social media studies. Social network analysis could replace isolated and static perspectives in studying interactive behaviors. Social network analysis uses a variety of mathematical techniques, such as maximum likelihood estimation and p-models, for studying stochastic and dynamic events to investigate the structural features of social media usage, which could be reflective of real contexts with certain levels of chaos, or entropy. Future works using social network analysis could use these techniques and corresponding hypotheses to investigate social media usage with a systematic and relative perspective, even in cooperation with other disciplines to address the complexity caused by entropy in systems. These techniques have the potential to profoundly reveal the underlying mechanism behind social media usage, so it is worthwhile for more social media studies scholars to pay more attention in the future.

**Author Contributions:** Conceptualization, Z.N.; methodology, Z.N.; writing—original draft preparation, Z.N.; writing—review and editing, M.W.; supervision, M.W., D.K. and W.A.B.W.A. All authors have read and agreed to the published version of the manuscript.

**Funding:** This research received no external funding.

**Conflicts of Interest:** The authors declare no conflict of interest.

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
