# Peer review of "The Role of Social Network Analysis in Social Media Research"

_applsci, doi:10.3390/app13179486_

Round 1

Reviewer 1 Report

The authors of this manuscript put forward a study that examines the impact of social networks on social media research. While the study is intriguing, there is a significant gap in the evaluation that is necessary to back up the hypothesis presented by the authors. Consequently, there are some significant concerns that need to be addressed.

* Major comments

In sections 2, 3, and 4 of the text, the authors conducted a thorough analysis. They examined how other authors used statistical methods to test their hypotheses, including models like linear regression (which seems to have some limitations in the analysis of this domain), entropy, and probabilistic equations. The text also discussed interesting features of the domain that should be analyzed, such as the interaction between different levels like micro and macro, users' behavioural patterns, and different types of communication between users. However, none of these features were explored in detail. 

Section 5 of the text presented several hypotheses that emerged from the study. However, two important points need to be addressed. First, it is not clear where these hypotheses were derived from. Second, hypotheses should be the starting point of the research, and a set of methods should be defined and designed to support or refute them. 

Overall, the study is quite comprehensive, including domain analysis and related work. However, the authors need to take the next step and provide values that represent their research objectives to materialize these hypotheses.   

* Minor comments

1. Introduction. At the end of the section, a brief remainder is needed to describe the content of the manuscript. 

* Spelling/grammar

4. The Method of Social Network Analysis. "environments. these dynamic" -> "environments. These dynamic". 

In conclusion, the manuscript presents an intriguing analysis of social networks and their influence on social media research. However, there are critical issues in the work that need to be addressed. I urge the authors to address the identified flaws.

The manuscript has been written in understandable English, and only minor changes are required.

Reviewer 2 Report

In this paper, the authors propose the use of social network analysis as a research method to address the limitations of previous studies in understanding social media usage patterns. By employing social network analysis, the authors aim to stimulate future investigations that delve into the complex nature of social media usage and its role within social networks.

However, some issues need to be addressed in the paper:

- There are certain sentences whose meaning is unclear, such as the following: "Prior research has predominantly relied on generalized concepts derived from established theories to examine the diverse usage patterns of social media. Nevertheless, the applicability of such concepts across different social contexts in real-life settings has yielded inconsistent findings. This inconsistency can be attributed to the fact that these concepts primarily reflect average levels of social media usage in general circumstances while failing to account for the dynamic and transient nature of social interactions",  and " These patterns of interaction offer insight into how small-scale interactions result in larger-scale social networks, as well as the characteristics of interactions, such as strong or weak ties, centrality structures, transition structures, etc.".

- Some statements require citations to support their claims. For example, statements like "Users spend an average of 6 hours and 35 minutes on social media, with connections with others being the most popular purpose for social media usage" and "Scholars typically find that while some factors have strong capacities to explain behavior in some contexts, these capacities were significantly attenuated in other circumstances, meaning that the factors could not always account for behaviors in all circumstances." should be accompanied by appropriate references.

- The paper lacks a Related Works section that cites relevant papers in the field of social network analysis and social media usage. It is recommended to include references to works that can provide insights into the application of social network analysis in studying common social networks [1, 2, 3].

- The contributions of the paper should be emphasized more clearly. It would be beneficial to explicitly state the novel aspects or insights.

- Figure 1 is difficult to read and should be improved for better clarity and understanding.

- The Conclusion section does not mention future works.

References

[1] Corradini, Enrico, et al. "Investigating negative reviews and detecting negative influencers in Yelp through a multi-dimensional social network based model." International Journal of Information Management 60 (2021): 102377.

[2] Corradini, Enrico, et al. "Investigating the phenomenon of NSFW posts in Reddit." Information Sciences 566 (2021): 140-164.

[3] Tabassum, Shazia, et al. "Social network analysis: An overview." Wiley Interdisciplinary Reviews: Data Mining and Knowledge Discovery 8.5 (2018): e1256.

Reviewer 3 Report

In general, the huge disadvantage of the present paper is constant lack of relevant sources when presenting new concepts, people`s habits etc. It needs in-depth review and real theoretical work with relevant references. Otherwise it could be considered as semi-scientific discourse.

In users’ habits add as up-to-date references as possible. They are constantly changing, and 5 years old study might not be relevant anymore.

Pg 1, Line 9: Just specify “prior research” on what?

Pg 1, Line 26-30: It is not clear whether the reference 1 is also reference for the statement in the first sentence. We do not see any comparison to other types of communication.

Pg 2, Line 55: List some scholars.

Pg 2, Line 60: Defining Social network analysis needs relevant references. There are none!

Pg 2, Line 85-87: Present some of those social contexts that could be “problematic” and add references.

Pg 2, Line 92: Briefly define transient social media usage, could be done in footnote. Add references.

Pg 3, Line 104: Only one study is not enough to be convinced. Add Pew RC studies, Reuters Institute etc.

Pg 3, Line 115-118: References on purposes and users’ habits in social media should be added.

 Pg 3, Line 122: More up-to-date references should be added.

Pg 3, Line 139-150: References are missing in the whole paragraph. In line 144 the Theory of reasoned Action and Theory of Planned Behaviour need to be referenced and explained.

Pg 4, Line 160: What studies? List & reference them.

Pg 4, Line 171: Add reference.

Pg 4, Line 180: How was it challenged? Explain through relevant studies.

Pg 4, Line 182-195: Add references on every step of incorporating methodological concepts, starting from the first sentence.

Pg 4, Line 200: It is not clear to what experiment this refers to. You need to describe it with all methodological aspects. “the behaviours we observed” is just a tease.

Pg 5, Line 214: Add another survey or more of them.

Pg 5, Line 236-238: List some of those empirical investigations and researchers.

Pg 5, Line 238-246: For those kind of observations references are mandatory. You cannot rely on general knowledge.

Pg 5, Line 254: No references next to definition?

Pg 6, Line 263-267: Reference missing.

Pg 6, Line 273-302: It is not clear whether the reference 24 refers to all 3 bolded sections. You cannot define Entropy without reference next to it.

Pg 6-7, Line 304-329: Not a single reference included, although many concepts are mentioned for the first time.

Pg 7, Lines 332-346: No references, even though mentioning concepts.

Pg 7, Line 349: What is Monte Carlo maximum? Where is the reference?

Pg 8, Line 380-383: Add references to traditional models?

Pg 10, Line 458: Which traditional theories?

Pg 10, Conclusion: Enrich the Conclusion with the most up-to-date researches relevant for your paper and compare them with your findings / set them into the context of your research.

Based on all abovementioned comments, unfortuntately, I cannot recommed the paper for acceptance.

Author Response

Thanks for the precious suggestions. Please see the attachment.

Round 2

Reviewer 1 Report

The authors have done incredible work, improving the quality of the work proposed by covering the gaps suggested. However, there are still some minor drawbacks that hinder the manuscript's comprehension.  

Minor comments. 

It is unclear how the formulas in the manuscript relate to the work's objective. For example, the authors use entropy to analyze the behaviour of multiple particles. However, it is unclear what the target of this comparison is and what hypotheses can be extracted for applying this formula to social networks in the domain of social media research. According to this definition, Equation 2 is used to estimate the probabilities of basic units of interactions. A social network with 5 actors is then employed to give a practical example of using this formula. The selected example is quite easy to follow. However, what are the implications of these computed estimations in the selected application domain? Then, the authors talked about relevant parameters like structure, topology, density and so on, but it is not clear what is the relation of these parameters with the study carried out. what are the targets or hypotheses expected to obtain, since the possible hypothesis shown in section 5, is neither related to these presented formulas nor talks about the parameters selected? For instance, how does the structure of the social network affect the final hypothesis obtained? and the topology? 

To conclude, while the work is interesting, there are still aspects that need improvement to enhance the quality of the manuscript.

Some spelling mistakes to review. 

* Spelling/grammar

Line 90. "The Significance of This Study" -> "The Significance of this study"

Line 195. " difficult it is to pipoint all" -> "difficult is to pinpoint?? all"

Line 230. "Not only is it impossible" -> This sentence is not clear. 

Reviewer 2 Report

The authors have addressed all of my concerns, and I believe the paper has improved since the last revision.

Author Response

Dear Reviewer:

Thanks for the comments, All the precious suggestions helped the refine of this paper, appreciation again.

All the best!

Reviewer 3 Report

The manuscript has been improved. However, some minor points remained:

- The text in lines 216-229, 310-340 and on other pages as well are bold. Correct.

- Future outlook is not sufficiently ambitious. Please, correct.

-Implications of findings for society should be elaborated.

- Fig. 1 is difficult to be read.

- In general, I suggest adding more references to your paper as it shows then that the topic is well-explored. At the moment, fresh references from 2022 onwards are quite poorly included. Please, improve.
